# Molecular detection and genetic characterisation of a large flood-borne outbreak of human leptospirosis in Jakarta, Indonesia: A retrospective analysis of surveillance data

Erni Juwita Nelwan[1,2‡], Yunita Windi Anggraini[2,3‡], Sabighoh Zanjabila[2,3], Budi Setiawan[4], Suhartiningsih Suhartiningsih[5], Farida Dwi Handayani[6], Jeny Jeny[3], Linda Erlina[7,8], Fadilah Fadilah[7,8], J. Kevin Baird[3,9], Raph L. Hamers[3,9], Suwarti Suwarti [2,3*]

1 Division of Tropical Medicine and Infectious Diseases, Department of Internal Medicine, Faculty of Medicine Universitas Indonesia/Cipto Mangunkusumo Hospital, Jakarta, Indonesia, 2 Infectious Disease and Immunology Research Center, Indonesian Medical Education and Research Institute, Faculty of Medicine, Universitas Indonesia, Jakarta, Indonesia, 3 Oxford University Clinical Research Unit Indonesia, Faculty of Medicine, Universitas Indonesia, Jakarta, Indonesia, 4 Jakarta Health Office, Jakarta, Indonesia, 5 Regional Health Laboratory, Jakarta, Indonesia, 6 Eijkman Research Center for Molecular Biology-Indonesian National Research and Innovation Agency, Cibinong Bogor Regency, Jakarta, Indonesia, 7 Bioinformatics Core Facilities, Indonesian Medical Education and Research Institute, Faculty of Medicine, Universitas Indonesia, Jakarta, Indonesia, 8 Department of Medical Chemistry, Faculty of Medicine, Universitas Indonesia, Jakarta, Indonesia, 9 Centre for Tropical Medicine and Global Health, Nuffield Department of Medicine, University of Oxford, Oxford, United Kingdom

‡ These authors contributed equally to this work and share first authorship.
* suwarti@oucru.org

## Abstract

Recurring outbreaks of leptospirosis in flood-prone areas caused by heavy rainfall pose a major public health concern, particularly in megacities such as Jakarta, Indonesia. From December 2019 through February 2020, Jakarta experienced a large leptospirosis outbreak due to extensive flooding following extreme monsoonal rainfall. We conducted a comprehensive retrospective analysis of the outbreak based on complete surveillance data from all five districts and 42 of 44 subdistricts in Jakarta. A total of 282 cases (97 suspected, 153 probable, and 32 confirmed) were reported in West (n = 162), South (n = 64), East (n = 30), North (n = 14) and Central (n = 12) Jakarta. Cases were predominantly adult males exposed to floodwaters. Of 241 cases tested, 164 (68.0%) had a positive IgM-based rapid diagnostic test (RDT). Of 118 cases tested with TaqMan RT-PCR targeting *lipL32*, 32 (27.1%) were positive. Of 95 cases tested with both assays, the combined detection rate was 74.7% (71/95); of whom 27 were positive by both RDT and RT-PCR. RT-PCR identified 5 additional RDT-negative cases, all of whom had fever <7 days. We sequenced 42 archived blood samples using Multi Locus Sequence Typing (MLST) and identified *Leptospira interrogans* and *L. borgpeterseni* as the predominant species. The findings emphasise the importance of rapid and early laboratory-based diagnosis during leptospirosis outbreaks in flood-prone urban areas, to better target public health interventions.

**Data availability statement:** All relevant data are within the manuscript and its Supporting information files.

**Funding:** The study was supported by an OUCRU Indonesia Young Scientist fellowship awarded to SS supported by The Wellcome Trust through Wellcome Africa Asia Programme, Vietnam (106680/Z/14/Z). JKB and RLH received salary from Wellcome Africa Asia Programme, Vietnam (106680/Z/14/Z). Additional support was provided by the Universitas Indonesia, through the PUTI Q1 2020 funding scheme (NKB-1301/UN2.RST/HKP.05.00/2020) awarded to EJN. The funders had no role in the study design, data collection, and analysis, decision to publish or preparation of the manuscript.

**Competing interests:** The authors have declared that no competing interests exist.

Climate-resilient urban planning is critical for vulnerable megacities in low-resource settings, where complex environmental and infrastructural challenges are compounded by the effects of a changing climate.

## Author summary

Leptospirosis is a bacterial disease transmitted through water contaminated by the urine of infected animals, such as rodents, livestock, or domestic animals. In flood-prone urban areas in low- and middle-income countries, heavy rainfall frequently spreads the bacteria widely in the environment, causing outbreaks. This study investigated a large leptospirosis outbreak following extreme seasonal rainfall from December 2019 through February 2020 in Jakarta, Indonesia. We identified 282 reported cases based on leptospirosis outbreak surveillance data, with adult males being the most affected by the outbreak through floodwater exposure. We evaluated the performance of leptospirosis rapid tests together with molecular detection, using RT-PCR and found that combining these methods improved early case detection. Genetic analysis showed that the outbreak was caused by two different species, named *L. interrogans* and *L. borgpetersenii*. This comprehensive analysis highlights the urgent need for improved diagnostic tests and surveillance systems, and enhanced disease control strategies. Climate-resilient urban planning is critical for vulnerable megacities where complex environmental and infrastructural challenges are compounded by the effects of a changing climate, including increased rainfall intensity.

## Introduction

Leptospirosis, a neglected zoonotic disease commonly found in tropical and sub-tropical regions, including Indonesia, is caused by spirochetes of the genus *Leptospira* spp., transmitted through water or soil contaminated by urine from infected livestock or rodents that serve as reservoirs [1,2]. Jakarta, Indonesia's capital and the world's most populous city (42 million), located on Java Island, frequently experiences flooding during prolonged heavy seasonal rainfall [3]. These seasonal floods, coupled with high *Leptospira* spp. carriage in sewer rats is a major driver of leptospirosis outbreaks [4]. Contributing environmental and infrastructural challenges include inadequate sanitation and drainage systems, low elevation worsened by rapid land subsidence, high population density and insufficient public health infrastructure. These are amplified by the emerging effects of climate change through rising sea levels, increased rainfall intensity and more frequent extreme weather events [5]. The end of December 2019 was the most significant rainfall in the last 24 years (reaching a high of 377 millimetres per day), causing recurrent floods up to the end of February 2020 [6]. On January 2, 2020, authorities declared an official public health alert following reports of human leptospirosis cases [7].

The Ministry of Health of Indonesia has implemented a surveillance system for the early detection of leptospirosis outbreaks, mainly relying on clinical case reports because of limited laboratory diagnostic capacity [8]. However, the diverse and non-specific clinical manifestations of leptospirosis lead to frequent misdiagnosis and underreporting [9]. Between 2019 and 2023, the annual number of leptospirosis cases ranged from 736 to 2554 nationwide and from 15 to 209 in Jakarta [10].

Although several surveillance studies have been conducted in Indonesia [11–13], there is a lack of laboratory-based case detection and confirmation during outbreaks. Reference diagnostic methods, such as the Microscopic Agglutination Test (MAT) and bacterial cultures, are expensive, require specialised expertise [14,15] and have limited availability in Indonesia [11]. Molecular diagnostic, Real-time PCR (RT-PCR) targeting the *lipL32* gene has been used in outbreak settings as a practical diagnostic tool for early detection and increased case identification [16,17], while Multi-locus sequence typing (MLST) is a useful method to identify pathogenic *Leptospira* species in clinical samples [16,17].

This study aimed to 1) describe the leptospirosis outbreak in Jakarta based on the complete government surveillance data from December 2019 through February 2020; 2) evaluate the performance of RDT and TaqMan RT-PCR *lipL32* as diagnostic tools during the outbreak; and 3) characterise circulating pathogenic *Leptospira* species using MLST.

## Methods

### Ethics statement

The study was approved by the Research Ethical Committee of the Faculty of Medicine Universitas Indonesia (19-05-0608) and the Oxford Tropical Research Ethics Committee (33–19). This study is reported as per the STrengthening the Reporting of OBservational studies in Epidemiology (STROBE) guidelines [18].

### Study design and study population

We performed a retrospective analysis of the complete surveillance data reported to the Jakarta Health Office from 21 December 2019 through 28 February 2020. Data were obtained from 24 primary healthcare centres, 16 public hospitals, and 15 private hospitals covering the five districts (West, South, East, North, and Central) and 42 of 44 flood-affected subdistricts. The two subdistricts not included, located in Kepulauan Seribu islands, did not report any leptospirosis cases.

The surveillance system recorded all suspected, probable, and confirmed leptospirosis cases, based on the case definitions of the Indonesian Ministry of Health [8]. According to the Ministry's surveillance guidance, serum samples were collected from all suspected leptospirosis cases for diagnostic tests (RDT and/or RT-PCR). Laboratory testing was performed at the healthcare facility (RDT) where available, or at the Regional Health Laboratory (RDT and RT-PCR). Suspected case was defined as acute fever (≥ 37.5 °C) with or without headaches accompanied by muscle pain, malaise, or conjunctival suffusion, and a history of exposure to risk factors (in this case, flood or water puddle) within the past two weeks. Probable case was defined as a suspected case with: i) two or more of the following clinical symptoms: calf pain, icterus, oliguria, haemorrhage, dyspnoea, arrhythmia, cough with or without haemoptysis, and skin rash; or ii) a positive RDT; or iii) at least three of the following laboratory results: thrombocytopenia (<100.000 cells/mm$^3$), neutrophilia (>80%), elevated total serum bilirubin, and proteinuria. Confirmed case was defined as a suspected case or probable case, with at least one of the following additional diagnostic test results: i) positive *Leptospira* culture from a clinical specimen; ii) positive PCR; iii) seroconversion from negative to positive or 4-fold titre rise from baseline to convalescent timepoint with MAT (not available during the outbreak).

### Data collection

A case record form captured case definition data, date of symptom onset, age, sex, flood exposure, fever onset, clinical symptoms, complete blood cell count and urinalysis. Leptospirosis diagnostic tests were conducted as part of surveillance

with IgM serological Rapid Diagnostic Test (RDT) (Pakar Biomedika, Indonesia) in each health care facility and TaqMan RT-PCR targeting the *lipL32* gene in the centralised Jakarta Regional Health Laboratory [8]. Daily precipitation data were obtained from the open data on the meteologix website [19], flood data from Indonesia's National Disaster Management Agency (BNPB) via Satudata [20], and district-level population data from BPS Statistics of Jakarta Province [21].

## Laboratory analysis

We conducted MLST and phylogenetic analysis of the residual whole-blood samples obtained for surveillance purposes and archived at the Jakarta Regional Health Laboratory. The total genomic DNA was extracted from 200μL whole blood using DNAeasy Blood and Tissue (Qiagen, Germany). MLST of the extracted DNA was performed through conventional PCR targeting *adk*, *icdA, lipL32, lipL41, rrs*, and *secY* genes in accordance with MLST scheme 3 [22]. All primers used were listed in S1 Table. PCR amplification was conducted for 40 cycles using GoTaq Green Master Mix in a 40 μL volume on a SimpliAmp Thermal Cycler (Applied Biosystem). Amplicons were visualised on a 2% agarose gel stained with SYBR Safe DNA gel stain (Invitrogen) and subsequently sequenced by First Base Laboratories (Selangor, Malaysia). All sequenced amplified molecular markers were concatenated. MLST allele and sequence types (STs) assignments were determined for samples with ≥4 positive loci, assigning them to the closest matching STs. Phylogenetic analysis of all sequences was performed using the Maximum-Likelihood tree construction method with 1000 bootstrap replicates and the General Time Reversible Gamma distribution Invariable sites (GTR + G + I) model, including reference sequences from the PubMLST database.

## Data analysis

Data analysis was conducted in R version 4.4.3 (R Core Team, Vienna, Austria). Data were presented as frequencies (%) and medians (IQR). District and subdistrict-level leptospirosis incidence rates were expressed per 100,000 person-years, calculated as the number of cases during the 3-month observation period divided by population size (December 2019 to February 2020). Geographic case mapping was plotted using QGIS version 3.32 with a basemap by the Geospatial Information Agency of Indonesia (https://geoservices.big.go.id). Test positivity and diagnostic yield of diagnostic tests were expressed as n/N (%), with a binomial 95% confidence interval (CI). The concatenation (cat), gene alignment (mafft), and multi-locus sequence typing (mlst) analysis were performed using a custom homebrew pipeline via command-line tools and analysed based on MLST scheme 3 from the PubMLST database (https://pubmlst.org/leptospira/). Species identification for the remaining samples that did not yield a complete MLST profile was conducted using BLAST analysis, comparing *secY* sequences, the most consistently amplified locus, against the NCBI database (https://blast.ncbi.nlm.nih.gov/) to confirm taxonomic identity. Complete allele assignment was not achieved, resulting in an incomplete MLST scheme 3 profile, the allele sequences could not be submitted to a PubMLST database. The phylogenetic tree was constructed in R version 4.4.3 (R Core Team, Vienna, Austria) [23] using Biostrings version 2.74.1 [24] to manage the sequence data, phangorn version 2.12.1 [25] for phylogenetic tree construction, and ape version 5.8.1 [26] for tree visualisation.

## Results

### Geospatial and temporal description of the outbreak

Fig 1 illustrates the outbreak timeline with daily reported cases, precipitation and the number of flood-affected subdistricts (S1 Fig presents weekly reports during the outbreak). The first probable case was reported on December 21, 2019 (week 1 of the outbreak). The outbreak reached its peak between January 10 and 14, 2020 (week 3–4), about two weeks after the first heavy flooding event (week 2). Subsequent floods occurred in the second half of January, 2020 (week 5–7), although with fewer subdistricts affected and fewer cases. More floods occurred in February (week 8–10), affecting a significant number of subdistricts but with few new cases. The last cases were reported on February 28, 2020.

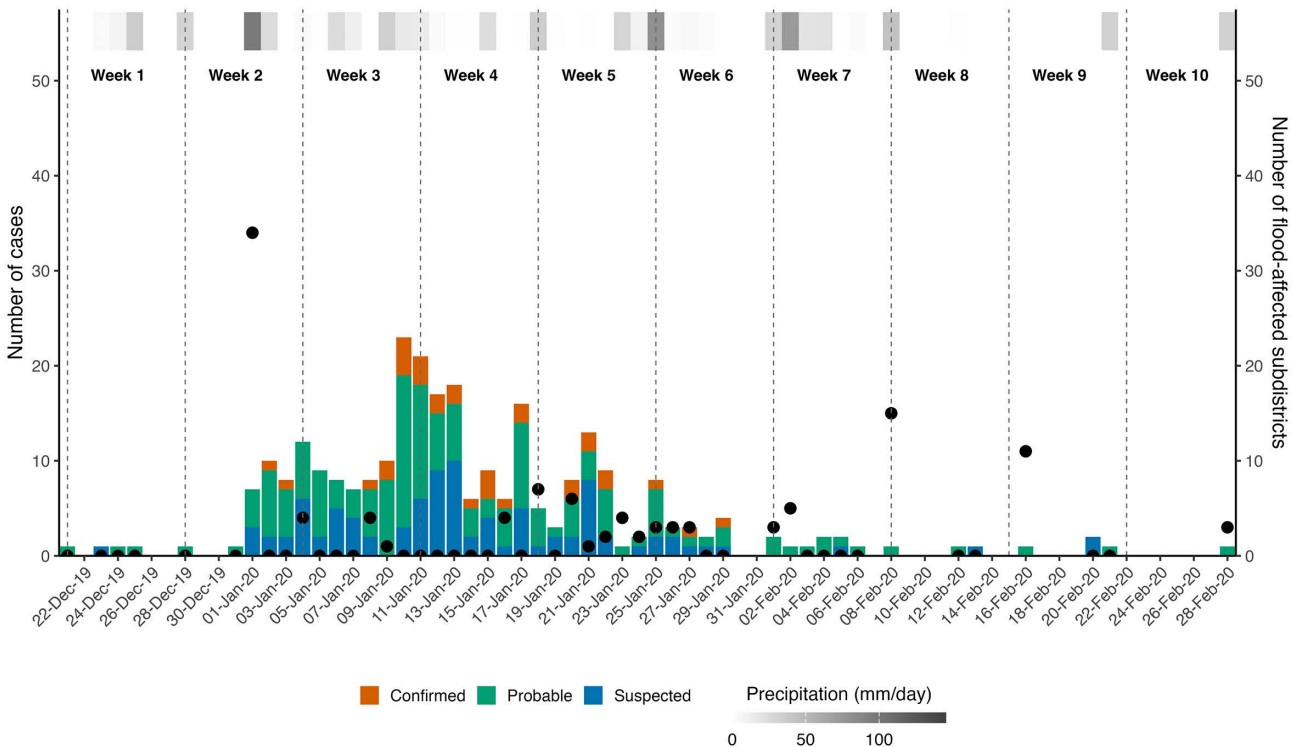

**Fig 1. The timeline of the leptospirosis outbreak in Jakarta, with daily reported cases, precipitation data and the number of flood-affected subdistricts from December 2019 to February 2020.** Bars represent suspected (blue), probable (green), and confirmed (red) cases recorded by the surveillance. The left y-axis expresses the total daily cases of leptospirosis (coloured bars), and the right y-axis represents the number of flood-affected subdistricts (black dots). The horizontal colour gradient at the top indicates daily precipitation in millimetres per day (mm/day).

All five districts of Jakarta were affected by the floods (35 of 42 [83.3%] subdistricts) and reported leptospirosis cases (33 of 42 [78.6%] subdistricts). The overall incidence rate in Jakarta was 10.7 per 100,000 person-years. West Jakarta had the highest leptospirosis incidence (n = 162; 26.6 per 100,000 person-years), with 6 of 8 subdistricts reporting cases (top-3: Cengkareng n = 72, Kalideres n = 23, Kebon Jeruk n = 22); followed by South Jakarta (n = 64; 11.5 per 100,000 person-years), across all 10 subdistricts (top-3: Tebet n = 16, Cilandak, and Kebayoran Baru n = 10 each); Central Jakarta (n = 12; 4.5 per 100,000 person-years), with 4 of 8 subdistricts reporting cases (top-3: Sawah Besar, Tanah Abang n = 4 each; and Kemayoran n = 2); East Jakarta (n = 30; 4.0 per 100,000 person-years), with 7 of 10 subdistricts reporting cases (top-3: Kramat Jati n = 10, Jatinegara n = 8, and Makasar n = 5); and North Jakarta (n = 14; 3.1 per 100,000 person-years), with 5 of 6 subdistricts reporting cases (top-3: Penjaringan, Tanjung Priok (a non-flooded subdistrict) each n = 4; and Cilincing n = 3). Notably, some non-flooded subdistricts, such as Tanjung Priok in North Jakarta and Tambora in West Jakarta, reported cases, while some flooded subdistricts, such as Kelapa Gading in North Jakarta and Cakung, Ciracas, and Cipayung in East Jakarta, did not report any cases (Fig 2 and S2 Table).

### Patient characteristics

Of the 285 cases, three cases had incomplete clinical and laboratory data; therefore, 282 cases were included in the analysis, as illustrated in Fig 3. The cases were classified as suspected (97, 34.4%), probable (153, 54.3%) or confirmed (32, 11.3%). Cases were predominantly male (n = 203, 72.0%), and aged 18–60 years (n = 212, 75.2%), and reported flood exposure (n = 228, 80.9%) within the preceding two weeks (Table 1).

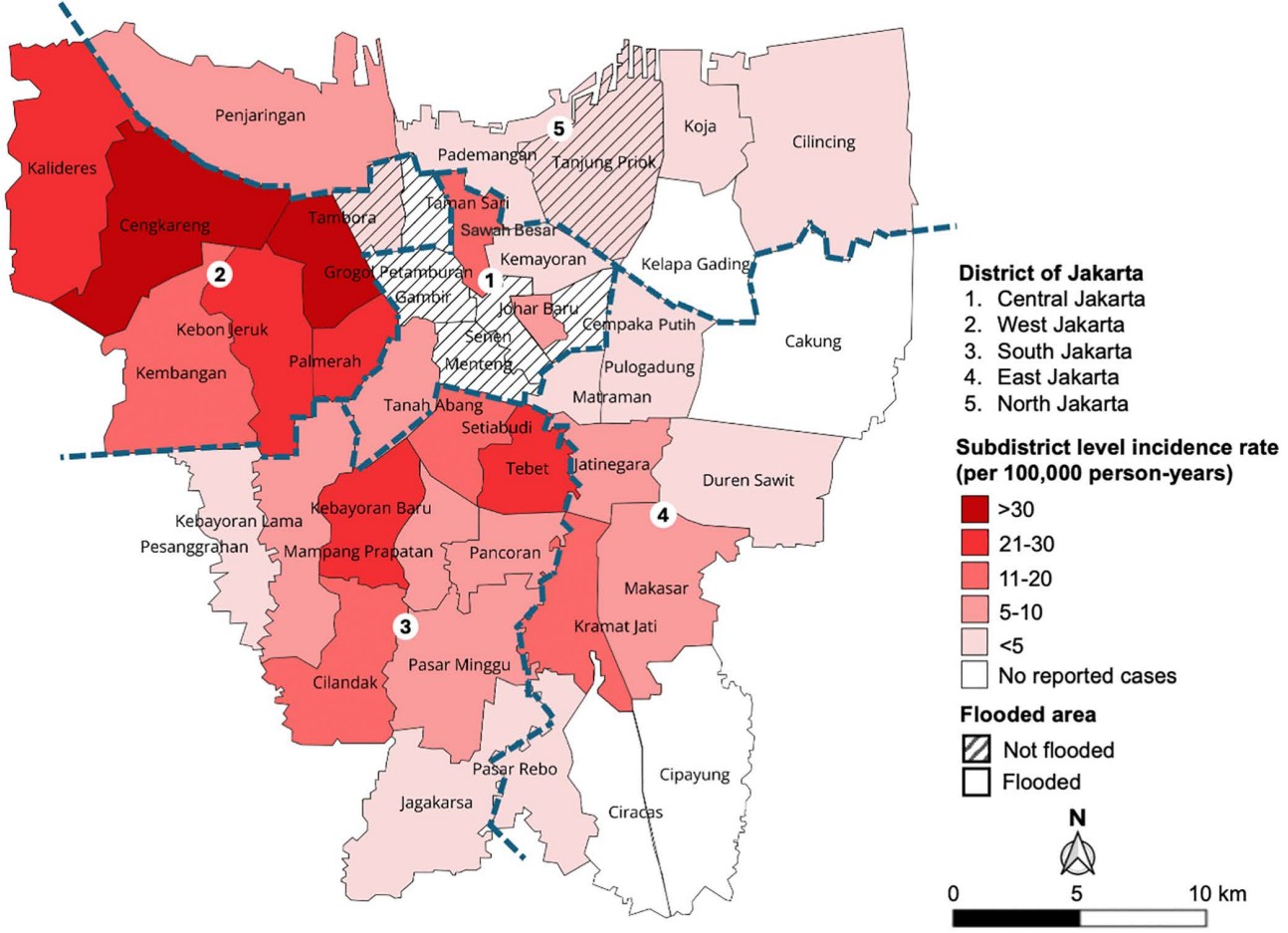

**Fig 2. Incidence rate of leptospirosis and spatial distribution of flood-affected subdistricts and districts from December 2019 through February 2020 in Jakarta.** The gradient from white to red represents the incidence rate, while not-flooded subdistricts are shaded. Maps were created in QGIS version 3.32 using basemaps from the Geospatial Information Agency of Indonesia (https://geoservices.big.go.id).

Table 1 summarises the patient characteristics. The most common characteristics were fever, myalgia, malaise, neutrophilia, calf pain, and conjunctival suffusion (Fig 4).

### Diagnostic test performance

Among the 282 cases included in the analysis, 241 patients were tested with RDT, of whom 68.0% (164/241) were RDT-positive. Of 118 tested with TaqMan RT-PCR *lipL32*, 27.1% (32/118) were positive. 95 (33.7%) of 282 cases were tested using both RDT and RT-PCR, while 18 (6.4%) were not tested by either method (Tables 1 and 2). RT-PCR detected an additional 5 cases who were RDT-negative (all of whom had fever 4–7 days), yielding a combined test positivity of 74.7% (71/95) (Table 2).

### *Leptospira* spp. identification and phylogenetic evaluation

PCR analysis on DNA from 42 archived whole-blood samples identified the *secY* gene in 15 samples (35.7%), *lipL32* in 13 (31%), *rrs* in 7 (1 6.7%), *adk* in 6 (14.3%), *lipL41* in 6 (14.3%), and *icdA* in 3 (7.1%). Overall, 17 samples (40.5%)

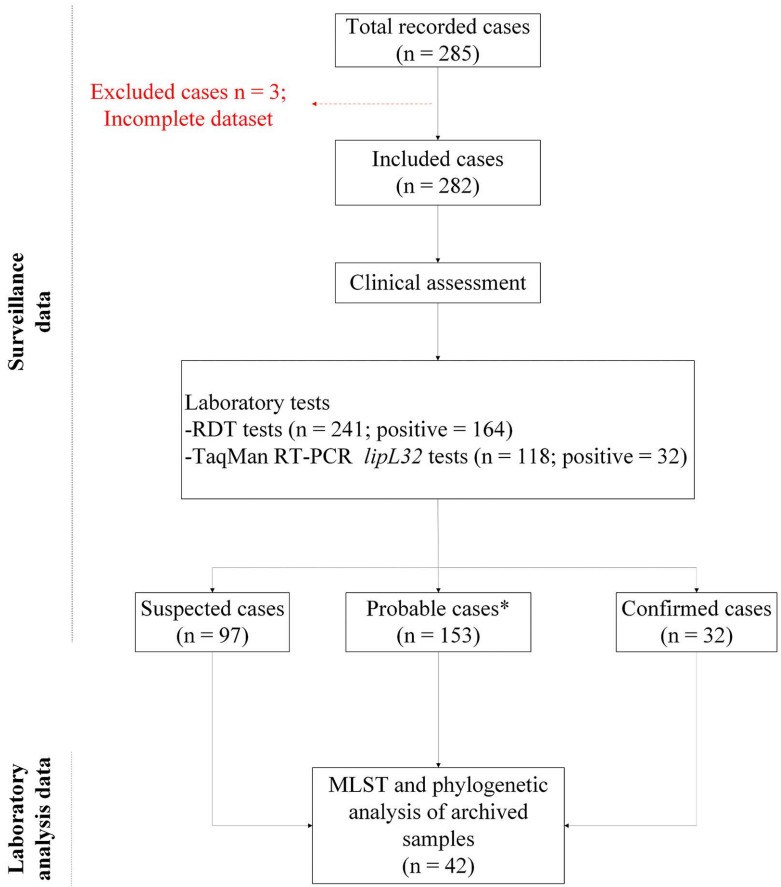

**Fig 3. Study flow diagram.** *) Probable cases based on Leptospirosis IgM RDT positives = 91.5% (140/153) and based on clinical symptoms combined = 8.5% (13/153). Abbreviations: MLST, multi-locus sequence typing; PCR, polymerase chain reaction; RDT, rapid diagnostic test; RT-PCR, real-time polymerase chain reaction.

were confirmed positive for leptospirosis through the detection of at least one of these MLST genes (S3 Table and S2 Fig). However, MLST scheme 3 was only able to assign the allele for 5 samples (01, 04, 14, 55, and 71) (Table 3). Among these five, samples 01 and 04 were identified as sequence types (STs) of *L. borgpetersenii* (STs 193 or 194), while samples 14, 55, and 71 clustered within the *L. interrogans* clade (Table 3 and Fig 5). Phylogenetic analysis indicates that *L. borgpetersenii* emerged around mid-January 2020 (week 4) in Central and North Jakarta, whereas *L. interrogans* was detected in West and Central Jakarta from mid-to-late January 2020 (week 6). Although the remaining 12 samples yielded incomplete MLST profiles, species-level identification was achieved through BLAST analysis of the *secY* locus, demonstrating >99% identity with *L. interrogans*, confirming that these isolates belong to the same species as the primary outbreak clusters.

## Discussion

To our knowledge, this is the first comprehensive molecular analysis of a large leptospirosis outbreak in Indonesia, identifying *L. interrogans* and *L. borgpetersenii* as the main circulating *Leptospira* species. Most (68%) cases were RDT-positive, and the addition of TaqMan RT-PCR *lipL32* increased case detection by 4.2%.

**Table 1. Patient characteristics and laboratory test results.**

| Characteristics | Total (%) | Cases (%) | | |
|---|---|---|---|---|
| | | Suspected | Probable | Confirmed |
| Number of cases | 282 (100) | 97 (34.4) | 153 (54.3) | 32 (11.3) |
| Age - Median (IQR) years | 38 (24–51) | 37 (24–49) | 39 (23–52) | 43 (33–55) |
| **Age group** | | | | |
| Children (1–12 years) | 23 (8.2) | 7 (7.2) | 14 (9.2) | 2 (6.3) |
| Adolescents (13–17 years) | 14 (5.0) | 5 (5.2) | 8 (5.2) | 1 (3.1) |
| Adults (18–60 years) | 212 (75.2) | 75 (77.3) | 112 (73.2) | 25 (78.1) |
| Elderly (>60 years) | 33 (11.7) | 10 (10.3) | 19 (12.4) | 4 (12.5) |
| **Sex** | | | | |
| Male | 203 (72.0) | 31 (32.0) | 43 (28.1) | 5 (15.6) |
| Female | 79 (28.0) | 66 (68.0) | 110 (71.9) | 27 (84.4) |
| **Occupation** | | | | |
| Employee/self-employee | 92 (32.6) | 31 (32.0) | 51 (33.3) | 10 (31.3) |
| Unemployed | 52 (18.4) | 17 (17.5) | 28 (18.3) | 7 (21.9) |
| Labourer | 50 (17.7) | 14 (14.4) | 28 (18.3) | 8 (25.0) |
| Housewife | 27 (9.6) | 8 (8.2) | 16 (10.5) | 3 (9.4) |
| Student | 21 (7.4) | 9 (9.3) | 12 (7.8) | 0 (0.0) |
| Unspecified | 40 (14.2) | 18 (18.6) | 18 (11.8) | 4 (12.5) |
| **District** | | | | |
| West Jakarta | 162 (57.4) | 60 (61.9) | 92 (60.1) | 10 (31.3) |
| South Jakarta | 64 (22.7) | 24 (24.7) | 33 (21.6) | 7 (21.9) |
| East Jakarta | 29 (10.3) | 8 (8.2) | 18 (11.8) | 3 (9.4) |
| North Jakarta | 14 (5.0) | 1 (1.0) | 6 (3.9) | 7 (21.9) |
| Central Jakarta | 13 (4.6) | 4 (4.1) | 4 (2.6) | 5 (15.6) |
| **Exposure to flood waters in the past 2 weeks** | | | | |
| Yes | 228 (80.9) | 73 (75.3) | 126 (82.4) | 29 (90.6) |
| No | 29 (10.3) | 11 (11.3) | 17 (11.1) | 1 (3.1) |
| Unspecified | 25 (8.9) | 13 (13.4) | 10 (6.5) | 2 (6.3) |
| **Leptospirosis diagnostic test *** | | | | |
| Rapid Diagnostic Test (RDT) | 241 (85.5) | 65 (67.0) | 147 (96.1) | 29 (90.6) |
| Positive | 164 (68.0) | 0 (0.0) | 140 (95.2) | 24 (75.0) |
| Negative | 77 (32.0) | 65 (100) | 7 (4.6) | 5 (15.6) |
| Not done | 41 (14.5) | 32 (33.0) | 6 (3.9) | 3 (9.4) |
| TaqMan RT-PCR *lipL32* | 118 (41.8) | 39 (40.2) | 47 (30.7) | 32 (100) |
| Positive | 32 (27.1) | 0 (0.0) | 0 (0.0) | 32 (100.0) |
| Negative | 86 (72.8) | 39 (100.0) | 47 (100.0) | 0 (0.0) |
| Not done | 163 (57.8) | 58 (59.8) | 105 (68.6) | 0 (0.0) |
| Both RDT and TaqMan RT-PCR *lipL32* | 95 (33.7) | 20 (20.6) | 46 (30.1) | 29 (90.6) |
| Positive | 71 (74.7) | 0 (0.0) | 42 (91.3) | 29 (100) |
| Negative | 24 (25.3) | 20 (100.6) | 4 (8.7) | 0 (0.0) |
| Unrecorded in any tests | 18 (6.4) | 13 (13.4) | 5 (3.3) | 0 (0.0) |

Data are n (%) unless specified otherwise.

*Includes all samples tested with the pertinent assay.

Abbreviations: IQR, interquartile range; RT-PCR, real-time polymerase chain reaction.

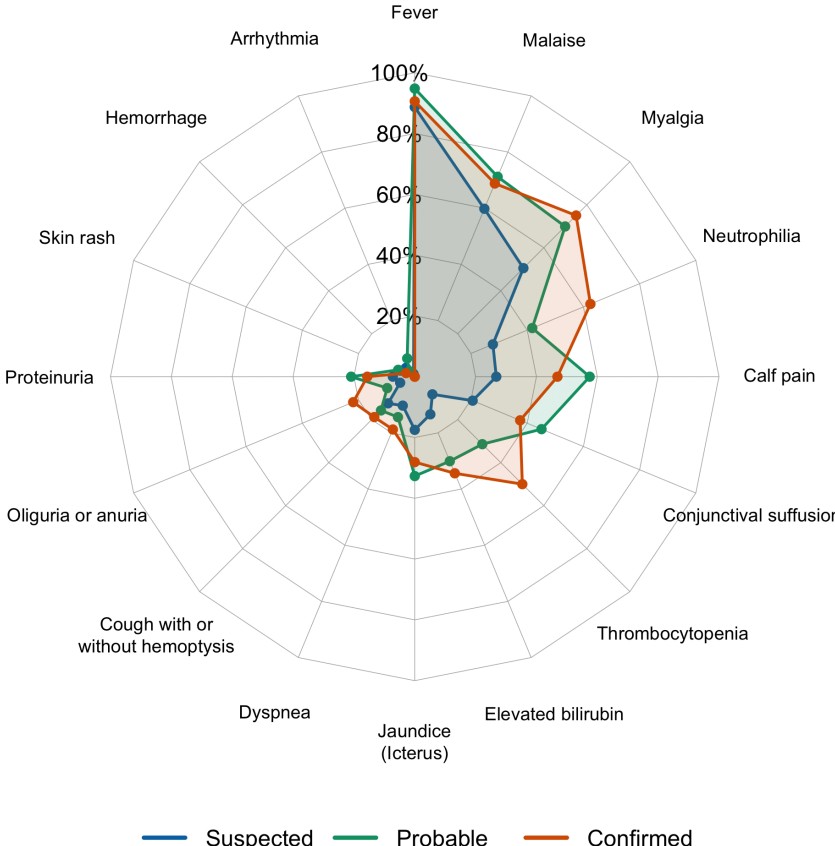

**Fig 4. Spider chart showing clinical characteristics of the 282 leptospirosis cases.**

**Table 2. Case detection by use of RDT and TaqMan RT-PCR *lipL32* for leptospirosis surveillance.**

| Diagnostic test (n = 95) * | Test-positive results | Test-negative results | Positivity rates % (95%CI) |
|---|---|---|---|
| RDT | | | |
| Fever onset 4–7 days | 22 | 29 | 43.1 (29.5-56.7) |
| Fever onset >7 days | 13 | 16 | 44.8 (26.7-62.9) |
| Fever onset unspecified | 7 | 8 | 46.7 (21.4-71.9) |
| TaqMan RT-PCR *lipL32* | | | |
| Fever onset 4–7 days | 5 | 46 | 9.8 (1.6-18.0) |
| Fever onset >7 days | 0 | 29 | 0.0 |
| Fever onset unspecified | 0 | 15 | 0.0 |
| Either RDT or TaqMan RT-PCR *lipL32* positive | | | |
| Fever onset 4–7 days | 39 | 12 | 76.5 (64.8-88.1) |
| Fever onset >7 days | 20 | 9 | 69.0 (52.2-85.7) |
| Fever onset unspecified | 12 | 3 | 80.0 (59.8-100.0) |

*Includes all 95 samples for which both RT-PCR and RDT test results were available.

Abbreviations: CI, confidence interval; RDT, rapid diagnostic test; RT-PCR, real-time polymerase chain reaction.

**Table 3. The result of MLST, allele assignation, presumptive STs and serovar of *Leptospira* in clinical samples.**

| Sample ID | Alleles assignation | | | | | | Presumptive STs* (MLST scheme 3) | Species | Presumptive serovar** |
|---|---|---|---|---|---|---|---|---|---|
| | adk | icdA | lipL32 | lipL41 | rrs | secY | | | |
| 01 | – | – | 10 | 15 | 30 | 49 | 193 or 194 | *L. borgpetersenii* | NA |
| 04 | – | – | 10 | 15 | 30 | 49 | 193 or 194 | *L. borgpetersenii* | NA |
| 14 | 5 | 2 | 2 | 50 | – | 13 | 154 | *L. interrogans* | NA |
| 55 | 5 | 1 | 2 | 3 | – | 5 | 149 | *L. interrogans* | Bataviae/Canicola |
| 71 | 5 | 1 | 2 | 3 | – | 5 | 149 | *L. interrogans* | Bataviae/Canicola |

- Indicates no match alleles.

* Samples with an allelic profile were assigned to the nearest match (≥4 matching loci) of sequence type (ST) in the *Leptospira* PubMLST database. (https://pubmlst.org/leptospira/).

* *The presumptive serovars were determined based on sequence type (ST) matching in the *Leptospira* PubMLST database.

Abbreviations: ID, identity; MLST, multi-locus sequence typing; NA, not applicable; ST, sequence type.

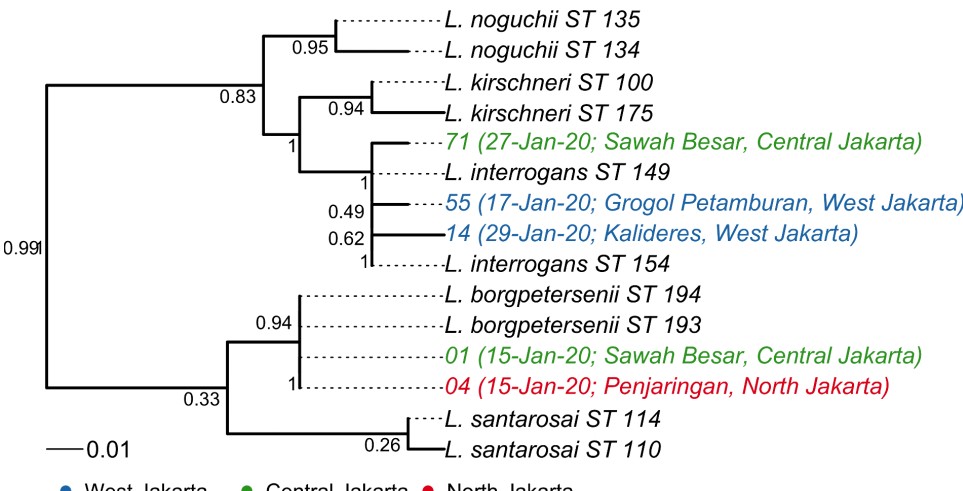

● West Jakarta   ● Central Jakarta   ● North Jakarta

**Fig 5. Maximum Likelihood tree of *Leptospira* using concatenated sequences from MLST scheme 3 loci gene sequences alignment, constructed under GTR+G+I substitution model.** Clinical samples (01, 04, 14, 55, and 71) are shown in relation to commonly reported reference pathogenic *Leptospira* sequence types (STs) from the PubMLST database. Bootstrap values (×1000) are displayed at the branch nodes to demonstrate the reliability of phylogenetic relationships. Corresponding sequence type isolate IDs and allele information from the PubMLST database are provided in S4 Table.

Flood-related leptospirosis outbreaks are common in tropical and subtropical regions, often driven by intense [27] or prolonged rainfall [28,29], facilitating waterborne transmission, compounded by environmental and infrastructural challenges and the effects of a changing climate [5]. West Jakarta reported the highest number of leptospirosis cases, which may be because the district is prone to extensive and long-duration flood events, explained by the district's topography, which includes proximity to the Java Sea, numerous lakes and rivers that induce severe flooding and slow water drainage during periods of heavy rainfall [30].

We observed an early peak in cases after heavy rainfall and severe flooding, followed by an extended outbreak with less precipitation and lower case numbers. This biphasic pattern suggests that the dissemination of *Leptospira* is likely associated with heavy precipitation and overflowing sewage systems, as reported in a previous study [31]. The time lag of 10–14 days between heavy flood events and the peak in cases could be due to a delayed recognition that an outbreak had started, alongside the incubation period of the infection. Consistent with this observation, outbreaks reported in other

settings occurred within 2–4 weeks post-flooding [32,33]. The predominance of middle-aged males is consistent with studies elsewhere, reflecting a higher risk of exposure to contaminated floodwaters [29,32,34]. Our findings underscore the importance of increased clinical awareness in flood-prone areas during the rainy season and access to rapid diagnostic tests for early outbreak detection, and prompt diagnosis and treatment [35].

The various diagnostic methods for leptospirosis each have strengths and limitations that influence their utility during outbreak investigations. *Leptospira* spp. culture is unreliable in outbreak settings due to slow bacterial growth, while MAT is limited to a few referral laboratories, further delaying diagnosis and treatment. Serology-based tests, like ELISA and RDT, require the accumulation of detectable amounts of anti-*Leptospira* antibodies, typically present in late acute to convalescent samples (10–14 days post-infection) [36]. Although currently available RDTs have a wide variation in reported sensitivity (68%–93%), they remain an essential tool for accessible, point-of-care diagnosis during outbreaks, particularly in settings with limited laboratory capacity [16,37]. Add-on RT-PCR targeting the *lipL32* gene has been reported to improve case detection and confirmation by 3–6% [38,39], particularly in the first 3–8 days after symptom onset [40]. We detected five additional cases out of 95 dual-tested (5.3% yield), all within seven days of symptom onset. However, RT-PCR is known to have reduced sensitivity after more than seven days of symptoms [41]. Moreover, variation in diagnostic yield across studies may reflect variability in patient populations, timing of sampling, sample quality, and assay performance, highlighting the need to further improve the implementation of current RT-PCR protocols. Taken together, the combined use of molecular and serological diagnostic tools is critical for improving case detection during outbreaks.

This study identified *L. interrogans* and *L. borgpetersenii* as the dominant pathogenic species, which are also the two most abundant species detected in rodent populations and human disease in Indonesia and Southeast Asia [42–44]. Most studies in Southeast Asia have focused on human infection in rural settings, like rice cultivation and other humid habitats. Our study highlights infection risks in flood-prone, densely populated urban environments, where *Leptospira*-carrying rats live in the sewage system [42]. In addition, *L. interrogans,* with serovars Bataviae and Canicola, have been found to be predominant in domestic animals [45], indicating that stray dogs and cats may serve as an additional reservoir during urban floods. We did not detect any other species, for example, *L. kirschneri*, *L. wolffii*, *L. santarosai*, and *L. weilii,* which have been previously reported in outbreaks in Southeast Asia [42,46], or *L. licerasiae*, an intermediate pathogenic species found in soil in Jakarta that can also infect humans during flooding events [47]. Our findings suggest that the outbreak caused by multiple *Leptospira* species, reflecting their co-circulation within the same environment, which may be due to their abundance in reservoir hosts, stagnant water and soil [44,47,48].

There are several limitations to this study. First, we relied on routinely collected surveillance data. This meant that some demographic, clinical, and laboratory data, including disease severity, were incomplete and that we could not assess patient outcomes. Some suspected cases were not laboratory-tested for leptospirosis, mostly due to logistical reasons (such as kit or reagent stockouts, sample not taken). Second, surveillance priorities shifted to the COVID-19 pandemic in March 2020, leaving the possibility that later cases were unrecorded. Third, we were not able to conduct leptospirosis reference testing, by culture and MAT, on the samples due to limited resources. Lastly, because we only genetically characterised a subset of the pathogens, we may have missed additional *Leptospira* species.

In conclusion, this analysis illustrates the risks of large leptospirosis outbreaks in vulnerable megacities where the complex interaction of infectious diseases, environment, infrastructure, and climate change presents formidable challenges. There is an urgent need for improved diagnostic tests and surveillance systems, enhanced disease control strategies and climate-resilient urban planning.

## Supporting information

**S1 Fig. The spatial distribution of total leptospirosis cases per week and flood-affected subdistricts between December 2019 and February 2020 in Jakarta.**
(TIF)

**S2 Fig. Maximum likelihood phylogenetic tree of *Leptospira* based on concatenated sequences of PCR-amplified MLST loci (S3 Table).** It illustrates strain diversity across subdistricts in Jakarta and the dates of sample collection. Phylogenetic inference was performed under the best-fit substitution model GTR + G (4) +I.
(TIF)

**S1 Table. List of primers used in this study.**
(XLSX)

**S2 Table. Geospatial distribution of the leptospirosis cases across the districts and subdistricts of Jakarta (December 2019 to February 2020).**
(XLSX)

**S3 Table. PCR-positive samples for *Leptospira* in six MLST loci.**
(XLSX)

**S4 Table. Reference isolates from the *Leptospira* PubMLST database used in phylogenetic tree construction, including sequence types (STs) and allele profiles based on MLST scheme 3.**
(XLSX)

## Acknowledgments

The authors express their gratitude to the Jakarta Health Office, the Ministry of Health of Indonesia, and the healthcare professionals who contributed to the data collection. The authors were also grateful to Kartika Saraswati, PhD and Made Ananda Krisna,PhD, for their valuable input during the development of this paper.

## Author contributions

**Conceptualization:** Erni Juwita Nelwan, Raph L. Hamers, Suwarti Suwarti.

**Data curation:** Budi Setiawan, Suhartiningsih Suhartiningsih.

**Formal analysis:** Yunita Windi Anggraini, Sabighoh Zanjabila, Linda Erlina, Fadilah Fadilah.

**Funding acquisition:** Erni Juwita Nelwan, J. Kevin Baird, Raph L. Hamers, Suwarti Suwarti.

**Investigation:** Yunita Windi Anggraini, Sabighoh Zanjabila, Jeny Jeny, Suwarti Suwarti.

**Methodology:** Yunita Windi Anggraini, Sabighoh Zanjabila, Farida Dwi Handayani, Jeny Jeny, Linda Erlina, Suwarti Suwarti.

**Project administration:** Sabighoh Zanjabila.

**Supervision:** Erni Juwita Nelwan, Suwarti Suwarti.

**Visualization:** Yunita Windi Anggraini, Sabighoh Zanjabila, Linda Erlina, Fadilah Fadilah, Suwarti Suwarti.

**Writing – original draft:** Yunita Windi Anggraini.

**Writing – review & editing:** Erni Juwita Nelwan, Farida Dwi Handayani, Suwarti Suwarti.

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
