## [Decision Letter · Decision Letter 0]

14 Nov 2025

PNTD-D-25-01675

Molecular detection and genetic characterisation of a large flood-borne outbreak of human leptospirosis in Jakarta, Indonesia: a retrospective analysis of surveillance data

Dear Dr. Suwarti,

Thank you for submitting your manuscript to PLOS Neglected Tropical Diseases. After careful consideration, we feel that it has merit but does not fully meet PLOS Neglected Tropical Diseases's publication criteria as it currently stands. Therefore, we invite you to submit a revised version of the manuscript that addresses the points raised during the review process.

Please submit your revised manuscript within by Jan 13 2026 11:59PM. If you will need more time than this to complete your revisions, please reply to this message or contact the journal office at plosntds@plos.org. Please include the following items when submitting your revised manuscript:

We look forward to receiving your revised manuscript.

Kind regards,

Claudia Munoz-Zanzi

Guest Editor

Elsio Wunder Jr

Section Editor

Shaden Kamhawi

co-Editor-in-Chief

Paul Brindley

co-Editor-in-Chief

**Additional Editor Comments (if provided):**

In addition to the reviewer's comments, please clarify:

- A description of the laboratory diagnostic effort during the period will provide more context for interpretation. Is it consistent across all districts? do all clinically suspect patients get lab tested with IgM and PCR or is it up to the clinician to request? Indicate the nature of the reporting to the MoH. Mandatory for all lepto categories?

- Figure 3 is confusing. The boxes from PCR, MLST are coming from Probable. It should show first boxes for the various tests and end with the case classifications.

- Probable cases: It seems most were IgM+ but the breakdown for the other 2 possible criteria should be clear because they are technically still clinically suspect only.

- Table 1: Methods include MAT but Table 1 does not show MAT results.

- Figure 5: The information is just a few numbers and they are already provided in the text. It appears that the figure is not necessary.

**Journal Requirements:**

**Reviewers' Comments:**

Reviewer's Responses to Questions

**Key Review Criteria Required for Acceptance?**

**Methods**

-Are the objectives of the study clearly articulated with a clear testable hypothesis stated?

-Is the study design appropriate to address the stated objectives?

-Is the population clearly described and appropriate for the hypothesis being tested?

-Is the sample size sufficient to ensure adequate power to address the hypothesis being tested?

-Were correct statistical analysis used to support conclusions?

-Are there concerns about ethical or regulatory requirements being met?

Reviewer #1: The objectives of the study are clearly articulated with a clear testable hypothesis, the design is appropriate to address the stated objectives; the population is clearly is described it is appropriate for the hypothesis being tested;

-Is the sample size sufficient to ensure adequate statistical power to address the hypothesis being tested;there are no concerns about ethical or regulatory requirements.

Reviewer #2: The objectives were clearly described. The study design was appropiate. The statistical analysis was correct. And there is no concerns about ethical issue. However, the concatenated MLST sequence alignment and phylogenetic analysis were not described throughly for exmaple R library used or software (which command) used. And these command and library should be cited properly.

**Results**

-Does the analysis presented match the analysis plan?

-Are the results clearly and completely presented?

-Are the figures (Tables, Images) of sufficient quality for clarity?

Reviewer #1: I have requested additional analysis and modification of the figures

Reviewer #2: The results were still not analyse throughly. There was no demonstration on the precipitation in sub-distrist to demonstrate the relation between (suspected, proable and confirmed) cases and precipitation, or between cases and flooded areas. If author can present in a series of weekly accumulate case numbers for each subdistrict, it will show how the outbreak occurred.

I suggust to show the border line of each five district on the Jakata Map to be clear for readers.

In Table 1: the Fever onset section, three groups (<4, 4-7, >7 ) might give more clear picture for concerdering performance of RDT and PCR in each group case.

Table 1 lacking outcome after treatment and the final diagnosis. This will be useful for investigate the performance of RDT as it could give a high false positive in some brand.

In Figure 4, it is better to used the differet color among the suspected, probable, and confirm groups in comparison

Authors can identify the Leptospira species from the amplified MLST loci and report in the Table 2 and further analysis.

For five MLST samples, two separate tree should be reconstructed based on 5 and loci, respectively.

**Conclusions**

-Are the conclusions supported by the data presented?

-Are the limitations of analysis clearly described?

-Do the authors discuss how these data can be helpful to advance our understanding of the topic under study?

-Is public health relevance addressed?

Reviewer #1: The conclusions are supported by data, the limitations are described. I have requested to add some comments about the potential use of these techniques for future studies.

Reviewer #2: It might be much better to see the more discussion on how the increase cases occured only after the first heavy rain (1 Jan 20) but not the later heavy rains (and after several floods).

Add more discussion on the RDT and PCR accuracy related to the fever onset

**Editorial and Data Presentation Modifications?**

Reviewer #1: no comments

Reviewer #2: (No Response)

**Summary and General Comments**

Reviewer #1: The manuscript “Molecular detection and genetic characterization of a large flood-borne outbreak of human leptospirosis in Jakarta, Indonesia: a retrospective analysis of surveillance data” describes leptospiral species and genetic variants in leptospirosis outbreaks associated with rainy and flooding periods in Jakarta-Indonesia, during 2019 and 2020. The study also evaluates a serologic IgM test and RT-PCR in the detection of leptospirosis cases. The results of this manuscript are important and timely.

Specific comments:

1) Lines 40-42. Improve the sentence for clarity

2) Lines 276-277. Discuss about the delay in leptospirosis cases after the rain, for example it seem unlikely that people are getting infected with rain water that washes off urine from soul.

3) Figure 6. Include information about date (months and years) and location, for sample sequences, in the phylogenetic tree. You could use branch color. Please discuss if there is any pattern.

4) Create phylogenetic trees with sequences from samples that produced less that 3 alleles in PCR, to visualize the diversity of leptospiral strains ( include dates in these trees) and place them as supplementary figures

5) Table 2. Please explain the logic for serovar determination (species determination many times is not correlated with serologic classification)

6) Discuss if there is clinical differences between L. interrogans and L. borgpetersenii infections.

7) Please discuss further if you recommend the use of serology IgM combined with RT-PCR to detect leptospirosis cases.

Reviewer #2: (No Response)

PLOS authors have the option to publish the peer review history of their article (what does this mean?). If published, this will include your full peer review and any attached files.

Reviewer #1: **Yes:** Gabriel Trueba

Reviewer #2: No

**Figure resubmission:**
---

## [Decision Letter · Decision Letter 1]

9 Mar 2026

PNTD-D-25-01675R1Molecular detection and genetic characterisation of a large flood-borne outbreak of human leptospirosis in Jakarta, Indonesia: a retrospective analysis of surveillance dataPLOS Neglected Tropical Diseases Dear Dr. Suwarti, Thank you for submitting your manuscript to PLOS Neglected Tropical Diseases. After careful consideration, we feel that it has merit but does not fully meet PLOS Neglected Tropical Diseases's publication criteria as it currently stands. Therefore, we invite you to submit a revised version of the manuscript that addresses the points raised during the review process. Please submit your revised manuscript by Apr 08 2026 11:59PM. If you will need more time than this to complete your revisions, please reply to this message or contact the journal office at plosntds@plos.org.  Please include the following items when submitting your revised manuscript:* A letter that responds to each point raised by the editor and reviewer(s). You should upload this letter as a separate file labeled 'Response to Reviewers'. This file does not need to include responses to any formatting updates and technical items listed in the 'Journal Requirements' section below.* A marked-up copy of your manuscript that highlights changes made to the original version. You should upload this as a separate file labeled 'Revised Manuscript with Track Changes'.* An unmarked version of your revised paper without tracked changes. You should upload this as a separate file labeled 'Manuscript'. If you would like to make changes to your financial disclosure, competing interests statement, or data availability statement, please make these updates within the submission form at the time of resubmission. Guidelines for resubmitting your figure files are available below the reviewer comments at the end of this letter. We look forward to receiving your revised manuscript. Kind regards, Elsio A Wunder Jr, DVM, Ph.D.Section EditorPLOS Neglected Tropical Diseases Elsio Wunder JrSection EditorPLOS Neglected Tropical Diseases

Shaden Kamhawi

co-Editor-in-Chief

Paul Brindley

co-Editor-in-Chief

 **Additional Editor Comments (if provided):**    **Journal Requirements:**

If the reviewer comments include a recommendation to cite specific previously published works, please review and evaluate these publications to determine whether they are relevant and should be cited. There is no requirement to cite these works unless the editor has indicated otherwise.**Reviewers' comments:** Reviewer's Responses to Questions

**Key Review Criteria Required for Acceptance?**

**Methods**

-Are the objectives of the study clearly articulated with a clear testable hypothesis stated?

-Is the study design appropriate to address the stated objectives?

-Is the population clearly described and appropriate for the hypothesis being tested?

-Is the sample size sufficient to ensure adequate power to address the hypothesis being tested?

-Were correct statistical analysis used to support conclusions?

-Are there concerns about ethical or regulatory requirements being met?

Reviewer #1: It was not clear to me whether they submitted the DNA sequences to GenBank or any other public database.

Reviewer #2: Please add a proper ciatation for R and R library used in the manuscript.

Redraft the author summary as it is currently as a short version of abstract.

**Results**

-Does the analysis presented match the analysis plan?

-Are the results clearly and completely presented?

-Are the figures (Tables, Images) of sufficient quality for clarity?

Reviewer #1: (No Response)

Reviewer #2: Table 1 -Should be clear in the last section that are the result of 95 samples, which be used for both PCR and RDT.

Table 2 -Please re-check the result of PCR as the toal should be 29 positive PCR

-Please remove every row of "Any fever onset"

-As there were <4 days of fever onset, I suggest group as i) 4-7 days fever onset and ii) >7 days fever onset

**Conclusions**

-Are the conclusions supported by the data presented?

-Are the limitations of analysis clearly described?

-Do the authors discuss how these data can be helpful to advance our understanding of the topic under study?

-Is public health relevance addressed?

Reviewer #1: The manuscript describes a large flood-borne (urban) outbreak of leptospirosis in Jakarta. The authors were able to obtain DNA sequences of leptospiral strains causing the outbreak. This is an important report because there is little information about these type of outbreaks.

Minor comments

1) In the discussion section it is necessary to address the evidence of different leptospiral strains causing infection in Jakarta form December 2019-February 2020. Please indicate whether other studies, in Indonesia or other parts of the world, have detected different strains affecting the same community at the same time. Also indicated the possible explanations for the diversity of leptospiral strains.

2) In discussion it will be interesting to know whether one os the leptospiral species caused more severe disease.

3) Fig S2 : The authors must indicate the genes used to build the tree. This figure shows that sequences from Jakarta fall in 3 clusters, 2 of them are identified as L. interrogans, the other one as L. borgpetersenii. The upper cluster remains unidentified, even though it is very conspicuous. I recommend to run Blast some of these sequences to see whether they belong to another leptospiral species.

Reviewer #2: (No Response)

**Editorial and Data Presentation Modifications?**

Reviewer #1: (No Response)

Reviewer #2: (No Response)

**Summary and General Comments**

Reviewer #1: (No Response)

Reviewer #2: After heavy rain and flooding, upserged suspected, probable and confirmed leptospirosis cases were reported. However, 11.3% were confirmed cases (32/282 positive PCR), while 49.6% were probable cases (140/282 positive RDT). It is common that RDT is not sensitive if the fever of onset less than 3-5 days. And it is common for false positive in the endemic area, tropical countries. The balance pro and cons of each assay use should be discuss and suggestion. When we should use PCR or RDT (single or pair sera) for giving more accurate early diagnosis & treatment or disease surveilance. What is the key message or lesson learn from this manuscript?

To demonstrate "a large flood borne" outbreak, the incidence of leptospirosis in west Jarkata are the highest in this study. However, authors did not compare to the incidence in the same duration in different years to point out the sharp increaseing of cases (suspected, probable or confirmed cases). Is topographic map can explain why the leptospirosis case number higher in the west than the other area.

It is interesting that more than one species of genus Leptospira identified. Author should more describe or discuss (or speculate) on relatedness of ST identified in the regions. For examples, the two close related isolate (55 and 71) of ST 149 found in west and central Jarkata, While two close related isolate of ST 193 found in north and central Jarkata. Interestingly, the central area, where the close related ST 149 and 193 identified, had few area effected by flooding.

PLOS authors have the option to publish the peer review history of their article (what does this mean?). If published, this will include your full peer review and any attached files.

Reviewer #1: **Yes:** Gabriel Trueba

Reviewer #2: No

  **Figure resubmission:** While revising your submission, we strongly recommend that you use PLOS’s NAAS tool (https://ngplosjournals.pagemajik.ai/artanalysis) to test your figure files. NAAS can convert your figure files to the TIFF file type and meet basic requirements (such as print size, resolution), or provide you with a report on issues that do not meet our requirements and that NAAS cannot fix.

After uploading your figures to PLOS’s NAAS tool - https://ngplosjournals.pagemajik.ai/artanalysis, NAAS will process the files provided and display the results in the "Uploaded Files" section of the page as the processing is complete. If the uploaded figures meet our requirements (or NAAS is able to fix the files to meet our requirements), the figure will be marked as "fixed" above. If NAAS is unable to fix the files, a red "failed" label will appear above. When NAAS has confirmed that the figure files meet our requirements, please download the file via the download option, and include these NAAS processed figure files when submitting your revised manuscript. **Reproducibility:** To enhance the reproducibility of your results, we recommend that authors of applicable studies deposit laboratory protocols in protocols.io, where a protocol can be assigned its own identifier (DOI) such that it can be cited independently in the future. Additionally, PLOS ONE offers an option to publish peer-reviewed clinical study protocols. Read more information on sharing protocols at https://plos.org/protocols?utm_medium=editorial-email&utm_source=authorletters&utm_campaign=protocols

---

## [Editor Report · Decision Letter 2]

9 Apr 2026

Dear Dr Suwarti,

We are pleased to inform you that your manuscript 'Molecular detection and genetic characterisation of a large flood-borne outbreak of human leptospirosis in Jakarta, Indonesia: a retrospective analysis of surveillance data' has been provisionally accepted for publication in PLOS Neglected Tropical Diseases.

Best regards,

Elsio A Wunder Jr, DVM, Ph.D.

Section Editor

Elsio Wunder Jr

Section Editor

Shaden Kamhawi

co-Editor-in-Chief

Paul Brindley

co-Editor-in-Chief

---

## [Editor Report · Acceptance letter]

Dear Dr -,

We are delighted to inform you that your manuscript, "Molecular detection and genetic characterisation of a large flood-borne outbreak of human leptospirosis in Jakarta, Indonesia: a retrospective analysis of surveillance data," has been formally accepted for publication in PLOS Neglected Tropical Diseases.

Best regards,

Shaden Kamhawi

co-Editor-in-Chief

Paul Brindley

co-Editor-in-Chief
